# Plants Saline Environment in Perception with Rhizosphere Bacteria Containing 1-Aminocyclopropane-1-Carboxylate Deaminase

**DOI:** 10.3390/ijms222111461

**Published:** 2021-10-24

**Authors:** Dhanashree Vijayrao Bomle, Asha Kiran, Jeevitha Kodihalli Kumar, Lavanya Senapathyhalli Nagaraj, Chamanahalli Kyathegowda Pradeep, Mohammad Azam Ansari, Saad Alghamdi, Ahmed Kabrah, Hamza Assaggaf, Anas S. Dablool, Mahadevamurthy Murali, Kestur Nagaraj Amruthesh, Arakere Chunchegowda Udayashankar, Siddapura Ramachandrappa Niranjana

**Affiliations:** 1Department of Studies in Biotechnology, University of Mysore, Manasagangotri, Mysore 570006, Karnataka, India; dhanashreebomle38@gmail.com (D.V.B.); ashkiran565@gmail.com (A.K.); jeevithakk98@gmail.com (J.K.K.); lavanyaraj16@gmail.com (L.S.N.); pradeep77.gowda@gmail.com (C.K.P.); 2Department of Epidemic Disease Research, Institutes for Research and Medical Consultations (IRMC), Imam Abdulrahman Bin Faisal University, Dammam 31441, Saudi Arabia; 3Laboratory Medicine Department, Faculty of Applied Medical Sciences, Umm Al-Qura University, Makkah P.O. Box 715, Saudi Arabia; ssalghamdi@uqu.edu.sa (S.A.); amkabrah@uqu.edu.sa (A.K.); hmsaggaf@uqu.edu.sa (H.A.); 4Department of Public Health, Health Science College Al-Leith, Umm Al-Qura University, Makkah 21961, Saudi Arabia; asdablool@uqu.edu.sa; 5Applied Plant Pathology Laboratory, Department of Studies in Botany, University of Mysore, Manasagangotri, Mysore 570006, Karnataka, India; botany.murali@gmail.com (M.M.); dr.knamruthesh@botany.uni-mysore.ac.in (K.N.A.)

**Keywords:** salinity, ACC deaminase, rhizosphere, rhizobacteria

## Abstract

Soil salinity stress has become a serious roadblock for food production worldwide since it is one of the key factors affecting agricultural productivity. Salinity and drought are predicted to cause considerable loss of crops. To deal with this difficult situation, a variety of strategies have been developed, including plant breeding, plant genetic engineering, and a wide range of agricultural practices, including the use of plant growth-promoting rhizobacteria (PGPR) and seed biopriming techniques, to improve the plants’ defenses against salinity stress, resulting in higher crop yields to meet future human food demand. In the present review, we updated and discussed the negative effects of salinity stress on plant morphological parameters and physio-biochemical attributes via various mechanisms and the beneficial roles of PGPR with 1-Aminocyclopropane-1-Carboxylate(ACC) deaminase activity as green bio-inoculants in reducing the impact of saline conditions. Furthermore, the applications of ACC deaminase-producing PGPR as a beneficial tool in seed biopriming techniques are updated and explored. This strategy shows promise in boosting quick seed germination, seedling vigor and plant growth uniformity. In addition, the contentious findings of the variation of antioxidants and osmolytes in ACC deaminase-producing PGPR treated plants are examined.

## 1. Introduction

The inability of plants to move from one place to another compared to other living organisms when subjected to environmental stresses can seriously impact them [1]. The environmental stresses experienced by plants are either biotic or abiotic. Biotic stresses occur when living organisms cause damage to the plants along with the deprivation of nutrients to the host, its inability to cope with such stress over time leads to plant death, hence resulting in pre- and postharvest loss [2,3]. Biotic stresses can occur due to bacterial, fungal and viral diseases, parasitism, physical damage by insects and nematodes, and competition and phytoparasitism by other plants [2,4]. Abiotic stresses, such as drought, water logging, salinity, extreme temperatures (cold, frost and heat), and heavy metal toxicity, are caused by non-biological components that are either environmental or nutritional, affecting the productivity of crops globally, thereby disturbing the plants’ growth and development (Figure 1) [5,6,7]. Of arable land, 90% is prone to these stresses resulting in yield losses of up to 70% in major food crops [8,9,10]. The constant exposure of plants to these abiotic stresses diminishes and limits crop yield. The stresses influence various plant responses such as changes in growth rates and crop yields, cellular metabolism and gene expression alteration, etc. Plants under stress mostly point to variations in their environmental conditions; the plant’s first response in such adverse conditions occurs in the roots [3]. A healthy and biologically diverse soil that tightly holds the plant increases the plant’s chances of surviving stressful conditions [11].

## 2. Events in Plant Salinity Stress; Interception by Rhizobacteria Containing ACC Deaminase

Among the abiotic stresses, salinity is considered to be one of the major environmental stresses that reduces crop yield worldwide [12]. The initial responses to salinity stress disrupt the Na^+^/K^+^ ratio in the plant cells’ cytoplasm [11]. It has been reported that in the past two decades, the degree of salinity has increased alarmingly by 37% in most irrigated land [13]. It affects the plants’ physiological and metabolic processes based on the severity and duration of the stress and results in low crop yield [14]. The saline soils cause osmotic stress and ion toxicity in plants, affecting their growth [15]. When plants experience any form of environmental stress, it results in increased ethylene formation, which is harmful to the plants, as in the case of salinity.

Plant growth-promoting microorganisms are a certain class of microorganisms that aid the plant’s growth either directly or indirectly. In contrast, the bacterial species aiding plant growth present at the rhizosphere are known as plant growth-promoting rhizobacteria (PGPR) [16]. The enzyme ACC deaminase produced by PGPR acts as a sink for 1-aminocyclopropane-1-carboxylic acid (ACC, which is an immediate biosynthetic precursor of ethylene), helping to lower the plant’s ethylene levels [17,18,19,20]. The metabolism of ACC by the action of this enzyme is the primary mechanism of PGPR to alleviate abiotic stresses, including salinity stress in plants [21]. The use of PGPR for aiding salinity stress tolerance is a highly beneficial green method that helps to increase plant growth and productivity [22]. Based on the importance of salinity, the present review summarizes (i) the effect of salinity stress in plants, (ii) how the plants try to withstand the salinity stress conditions, (iii) how the ethylene concentration increases during salinity stress and (iv) the mechanism of action through which PGPR alleviate salinity stress by producing ACC deaminase enzyme.

## 3. Altered Responses in Plants Due to Salinity and Its Consequences

During salinity stress, plants try to adapt to conditions by altered mechanisms at the cellular as well as the whole plant level [23]. The mechanisms at the cellular level are (i) regulation of osmotic potential, (ii) cell wall alteration, (iii) ROS elimination, (iv) vesicle transport, (v) transport protein generation, (vi) K^+^ and NO^3−^ homeostasis, (vii) distribution to vacuoles, and (viii) solute production. In contrast, at the whole plant level, the tolerance mechanisms include (1) the change in flowering and fruiting times and the re-translocation of photosynthates, (2) the distribution of leaf salts to the sheath/petiole rather than to the lamina or salt allocation to epidermis cells instead of mesophyll, or its excretion [24], (3) controlling the long transport and restoration of salt in the stem, (4) change in the structure of root, (5) the excretion of more than 95% of the root salt to the soil. (6) removal of excess salts from the xylem, and (7) the development of symbiotic associations with the soil microbes such as arbuscular mycorrhizal fungi and PGPR rhizospheric bacteria. 

A plant’s response to salinity stress will result in the production of stress genes, which will improve plant resistance to stress. In stress, epigenetic mechanisms regulate gene expression, which include changes in DNA and RNA activity and chromatin modification [25,26]. Changes in membrane structures caused by salt stress create metabolic stress, produce reactive oxygen species, and impede photosynthesis, resulting in nutrient deficit [27,28]. Signaling chemicals including nitric oxide (NO), hydrogen sulfide (H_2_S), H_2_O_2_, calcium, reactive oxygen species (ROS), and plant growth regulators, salicylic acid (SA), jasmonic acid (JA), ethylene (ET), and abscisic acid (ABA), all play important roles in cell signaling and crosstalk, allowing the cells to withstand a variety of stressors [29,30]. The adverse effects of sodium (Na^+^) and chloride (Cl^−^) ions on cellular homeostasis, as well as the osmotic potential, impacting plant physiology, particularly water intake, are the stressful consequences of salinity on plant growth [31,32]. Several homeostasis-signaling pathways are activated when plants are exposed to stresses such as salinity and drought. ROS also function as signaling products in plants, altering their stress tolerance. Plant tolerance to stress may be increased if the ROS signaling pathway response is at the appropriate level [33].

Salinity causes osmotic stress, which reduces leaf turgor by impeding water flow across the plant and reduces stomatal conductance (Gs) by closing the stomata [34,35]. The stomatal closure reduces the rate of transpiration (E) and the amount of CO_2_ available in the leaves. As a result, the intercellular CO_2_ concentration (Ci) decreases, causing changes in the leaf biochemistry that have a negative impact on the net CO_2_ assimilation rate (A) under long-term stress [34,36]. Salinity stress affects the various qualitative and quantitative parameters as well as major physiological processes in plants that are key factors to their existence and survival in the following ways:

### 3.1. Implications on Plant Growth and Development

When diploid *Robinia pseudoacacia* were treated with 250 mM NaCl, wilting and chlorosis were observed in the leaves and most of them etiolated from the leaf apex, whereas growth inhibition and damage to leaves were noticed at higher concentration (500 mM) of NaCl in the diploid and tetraploid *R. pseudoacacia*, respectively [37]. In the study on carnations subjected to salinity stress, the leaf area was reduced by 58% at 200 mM NaCl compared to the untreated control. When carnations (*Dianthus caryophyllus*) were subjected to salinity stress at the flowering stage, a drastic decrease in plant height by 68% compared to the untreated plants was observed at 200 mM NaCl concentration [38]. In Koroneiki olive cultivars exposed to 100 mM and 200 mM NaCl concentrations, the dry weight of roots was reduced by 46 and 67%, respectively. Such negative effects were also observed in the dry weight of shoot and stem, where they declined by 91% and 77%, respectively, in Koroneiki olive cultivars subjected to 200 mM NaCl. Moreover, leaf dry weight was reduced by 58 and 69% at 100 mM and 200 mM, respectively [39].

There were slightly different correlations between salinity stress and root dry weight in studies on carnations subjected to salinity stress. Though the dry weight of the root did not show any change, there was a significant decline in the stem’s dry weight when the carnation was subjected to salinity stress, thus resulting in a reduced shoot/root ratio [38]. When different sorghum genotypes under salinity stress were assessed and compared to the control, i.e., non-saline condition, the genotype PAYAM showed the maximum decrease in various germination parameters except mean germination time (MGT) [40]. The germination percentage (GP) and germination index (GI) decreased drastically in all sorghum genotypes subjected to salinity stress with the maximum decrease in GP and GI by 57 and 50%, respectively, at 200 mM NaCl concentration. The seedling vigor index recorded a decline in the PAYAM genotype. MGT was the other seedling parameter mainly affected, which showed a significant increase in various sorghum genotypes at 100 mM, 150 mM, and 200 mM NaCl. The maximum MGT observed in the PAYAM genotype was 60% at 200 mM NaCl compared to the control [40]. When the flowers produced by carnations subjected to salinity stress were monitored and assessed qualitatively and quantitatively, it was found that the weight and quality of the flowers decreased with increasing NaCl concentrations. The plants also failed to flower when grown at high concentrations (100 mM and 200 mM) of NaCl [38].

### 3.2. Effect on Plant Biomass and Yield

At 25 °C, saline soil has an electrical conductivity (EC) value of the saturation extract (ECe) in the root zone that exceeds 4 dS m^−1^ (about 40 mM NaCl) and an exchangeable sodium level of 15% [41]. At this value of ECe and below in several plants, the cultivation of crops leads to stunted growth, resulting in decreased crop production [42]. The phenomenon which primarily affects the plants because of the saline soil is the inhibition of seed germination, which drastically reduces the crop yield [43]. The salinity stress also leads to ionic toxicity, which disturbs the plant’s osmosis and unbalances the nutrient channels [44]. The disturbance in these important processes results in altered metabolism and physiology, ultimately leading to adverse effects on seed germination and seedling growth. In addition, the salinity stress has a deleterious impact on various metrics, including shoot and root lengths, fresh and dry weights, and plant chlorophyll synthesis retardation [44].

Moreover, this stress remarkably lowers the rate of seed germination in various plants [45]. In other words, the total biomass and yield of the plant is significantly decreased due to salinity stress which acts as an indicator of the ability of plants to withstand salinity stress [46]. The calculation consists of quantification of the net assimilation rate (NAR), leaf area ratio (LAR) and relative growth rate (RGR) of the whole plant and root system [47]. The leaf area ratio (LAR) is the ratio of leaf area to leaf dry mass. Relative LAR (RLAR) is the LAR measured under salinity stress condition compared to control condition, which gives an idea of the effect of this stress on the leaf thickness. The decrease in RLAR might be adaptive in salt conditions because the leaf has thicker cell walls or a larger volume into which salts can be sequestered [48].

The total leaf area has been observed to decrease under salinity stress conditions because of altered cell wall characteristics, changed leaf turgor pressure and a reduced photosynthesis rate [49]. The research on pearl millet has shown that the studied plant’s growth, productivity, and biomass were negatively impacted, particularly seed germination percentage, plant height, leaf area, total biomass, and crop yield [50]. The study on pea plants also revealed that salinity stress reduced its growth and yield, and ultimately its biomass. Furthermore, the studies conducted to understand the effect of salinity stress on various grain legumes indicate a reduction in crop yield in the range of 12–100% [43]. Different plant species have shown a higher dry root mass than shoot dry mass under high salinity, resulting in a higher root to shoot ratio, which is expected to improve the water and nutrient source/sink ratio under such conditions [49].

Salinity impacts flowering and fruiting patterns and reproductive physiology, which ultimately influences the agricultural yields and biomass. It has resulted in up to 50% decrease in the flowering of pigeon pea [43] and has also slowed wheat reproductive growth by limiting spike development, whereas in rice, it has reduced tillers and the formation of sterile spikelets, hence resulting in the reduction in yield [51].

### 3.3. Effect on Photosynthesis in Plants

In higher plants, photosynthesis occurs in chloroplasts, where a series of oxidation−reduction processes occur as part of light and dark reactions, converting light energy into chemical energy [52]. Light energy is used to produce ATP and NADPH and the release of oxygen during the light reaction. The ATP and NADPH produced during light reactions are used in dark reactions to fix CO_2_ in carbohydrates. Salinity stress affects the major physiological and metabolic processes of photosynthesis by increasing Na^+^ and Cl^−^ in leaf tissue of plants, which are considered stomatal, non-stomatal or both factors of photosynthetic limitation [53]. An increase in NaCl concentration significantly impacted stomatal closure, enzyme activity, pigment concentrations, metabolites, and the ultrastructure of various organelles such as chloroplasts, resulting in a reduced quantum yield of photosystems and a negative impact on photosynthetic efficiency [38,39,53]. It has also been observed that an overabundance of ROS negatively affects plant growth and production by promoting photo inhibition of PSII (photosystem II) and triggering degradation of photosynthetic pigments, oxidation of lipid molecules, and suppression of gene expression [54,55].

Sugar beet is considered a promising crop for cultivation in the saline-prone, large coastal regions in Bangladesh, as it can survive at low (75 mM NaCl) to moderate (100 mM NaCl) salinity as assessed by (i) the abundance of relative water content, which increased the succulence of the sugar beet, (ii) elevated photosynthetic pigment, such as chlorophyll content, (iii) balanced osmolyte content, such as proline, (iv) increased the net CO_2_ assimilation rate, stomatal conductance, carboxylation efficiency, and water use efficiency which contributed to better carbon and mineral management and, (v) increased antioxidant enzyme, such as catalase, ascorbate peroxidase, and peroxidase activity, which detoxifies excess reactive oxygen species with 150 mM NaCl stress, gas exchange parameters, and chlorophyll content were slightly hampered [56].

#### 3.3.1. Effect on Chlorophyll

Salinity induces a change in the amount of leaf chlorophyll due to impairment in the processes related to the pigment’s biosynthesis and an increase in the pigment’s degradation [57,58,59]. A drop in the concentration of precursors like glutamate and 5-amino levulinic acid in sunflower callus exposed to this stress, indicated that chlorophyll production was substantially hampered [53,60,61]. When *Elaeagnus angustifolia* leaves were subjected to increasing NaCl concentrations over a 7-day period, a drastic decrease in the contents of chlorophyll a, chlorophyll b, and the overall amount of chlorophyll was observed [62]. Transcriptome profiling indicated that seven differentially expressed genes (DEGs) were involved in the chlorophyll metabolism pathway in *E. angustifolia* seedlings grown under salinity stress. The DEGs encoding protochlorophyllide reductase A, protochlorophyllide reductase C, magnesium chelatase subunit chlD, magnesium chelatase subunit chlI, and protoporphyrinogen oxidase were downregulated after treatment with NaCl [62]. In research on the effect of salinity stress, the amount of chlorophyll was higher in salt-tolerant plant species, whereas salt-sensitive plant species had a drop in pigment content. However, the accumulation of this pigment is not always linked to salt tolerance, as various other studies have shown. For example, an experiment on tomato cultivars with varying salinity tolerance revealed a very weak relationship between the concentration of Na^+^ and chlorophyll, implying that the ability to tolerate salinity varies with plant species [63].

#### 3.3.2. Effect on Stomatal Regulation, Associated Gas Exchange Properties and Enzymes Involved in Photosynthesis

The osmotic effect induced by salinity stress causes higher abscisic acid (ABA) levels within stomatal guard cells, resulting in stomatal closure and a decrease in stomatal conductance, as well as a decrease in CO_2_ concentration, indicating lower photosynthesis [38,39,53,62]. The stomatal closure limits leaf CO_2_ assimilation by salinity stress which leads to the altered expression of genes encoding the key enzymes (such as Ribulose 1,5-bisphosphate carboxylase/oxygenase, sucrose phosphate synthase, nitrate reductase, phosphoenolpyruvate carboxylase, and pyruvate orthophosphate dikinase) involved in the process of carbon fixation [34,53,62]. When the carboxylate enzyme Ribulose 1,5-bisphosphate carboxylase/oxygenase is deprived of CO_2_, it inhibits the carbon dioxide fixation which has been observed bothin vivoandin vitrowhen exposed to high salt concentrations [64]. This enzyme prefers O_2_ as a substrate instead of CO_2_ as it is not in adequate amounts, resulting in the generation of reactive oxygen species (ROS) such as superoxide radical (O_2_**^.−^**), singlet oxygen (^1^O_2_), and hydrogen peroxide (H_2_O_2_) [54,65]. Another study found that Ribulose 1,5-bisphosphate carboxylase/oxygenase enzyme activity and protein contents were only 61 and 70%, respectively, in *E. angustifolia* seedlings under salinity stress, compared to those in the control, demonstrating how it negatively affects photosynthesis [62]. The activity of enzymes involved in Ribulose 1,5-bisphosphate carboxylase/oxygenase regeneration, such as ribulose-1,5-bisphosphate (RuBP), and fructose-1,6-bisphosphatase implicated in RuBP regeneration, was found to be downregulated similarly under salinity stress [53,62].

It has been noted that, increased Na^+^ accumulation disrupts electron transport and photosystem assembly, resulting in a decrease in the ATP and NADPH produced [52,53]. The reduction in the amount of ATP produced reduces the extent of RuBP regeneration and photosynthesis rates [34]. Accordingly, various additional genes for enzymes involved in dark reactions were downregulated in *E. angustifolia* seedlings during photosynthesis, such as fructose-1,6-bisphosphatase which converts fructose-1,6-bisphosphate into fructose-6-phosphate as a part of Calvin cycle, triosephosphate isomerase which catalyzes the inter-conversion of dihydroxyacetone phosphate and D-glyceraldehyde-3-phosphate (isomers of triose phosphate), malate dehydrogenase which catalyzes the reversible conversion of malate into oxaloacetate, and ribose-5-phosphate isomerase which catalyzes the interconversion of D-ribose-5-phosphate and D-ribulose-5-phosphate [62]. Other enzymes that require K^+^ as a cofactor were also significantly harmed since they were found to be affected by elevated Na^+^ levels or the Na^+^/K^+^ ratio [34].

With high levels of NaCl, the amount of K^+^ in the leaves and roots of *E. angustifolia* was significantly reduced, with the lowest amount of K^+^ found in the roots, followed by expanded leaves and expanding leaves [62]. At Na^+^ concentrations of more than 100 mM, the normal functioning of photosynthetic enzymes was found to be affected [34]. Due to the lack of carbon as a substrate, the diffusion of gases such as CO_2_ through stomata and leaf mesophyll decreases, lowering mesophyll conductance and impacting numerous genes [34,53]. As a result of this, the genes involved in photosynthesis and chloroplast folding are frequently downregulated. It is well documented that salt stress, which occurs when dehydration is combined with osmotic stress, affects a larger number of genes [34] which was substantiated in salt-sensitive sweet sorghum species, wherein the expression of genes involved in photosynthesis was downregulated [52].

#### 3.3.3. Effect on Photosystems

Salinity stress affects the pigments in the photosystems involved in photosynthesis, lowering the light-absorbing efficiency of photosystems I and II, and lowering photosynthetic capacity [66]. The variable/maximum fluorescence (F_v_/F_m_) ratio depicts the maximal quantum efficiency of photosystem II, which is a measure of overall photosynthetic capability, as a linear electron transport rate [67]. In healthy leaves of most plant species, the F_v_/F_m_ ratio is normally near 0.8; if it is less than 0.8, it indicates the presence of photo inhibition, a process found in stressed plants where the photosystem II centers have either been destroyed or inactivated [68]. When *E. angustifolia* seedlings were exposed to increasing concentrations of NaCl, the F_v_/F_m_ ratio dropped dramatically and the expression of DEGs linked with photosynthetic-antenna proteins encoding Lhca3, Lhcb1, Lhcb3, Lhcb4, and Lhcb6 was found to be downregulated, thereby confirming that the exposure of plants to high salt concentration disrupts the formation of absorption, transfer, and distribution of light energy [62].

### 3.4. Effect on Organelles

The light and dark reactions, that makeup photosynthesis occur in the chloroplast, a salt-stressed organelle [53]. Salinity stress causes an increase in sodium and chloride ions, which irreversibly damage the thylakoid membranes, impairing electron transport and photophosphorylation. Severe damage was detected in the ultrastructure of chloroplasts exposed to salinity stress in diploid black locusts, a sensitive rice variety (Amber), and pea cultivars [37,69,70]. It was observed that when the plants were exposed to high salt concentrations, there was swelling in the grana and thylakoids, a reduction in the number of grana, and a noticeable shift in the form of chloroplasts from ellipsoidal to oval. Disorganized chloroplast membranes and increased plastoglobuli levels, both in terms of quantity and size, were also noticed, due to thylakoids breaking down due to salinity [70,71]. Apart from the chloroplast, when the leaves of a salt-sensitive plant were compared to the leaves of a salt-resistant pea plant, the former had fewer cristae and lower mitochondrial electron density [70].

### 3.5. Effect on Hormone Production

Hormones play a crucial role in plant growth and development, and they are also effective against various stress conditions, including both biotic and abiotic stress factors [72]. It is well-known that plant hormones are produced at the commencement of stress and play a critical function in maintaining plant growth and development throughout stressful situations. In response to salt stress, a variety of phytohormones such as ABA, jasmonic acid (JA), gibberellic acid (GA_3_), ethylene, and salicylic acid are integrated and coordinated [73].

#### 3.5.1. Effect on Abscisic Acid (ABA) Production

Abscisic acid has been found to govern seed germination and subsequent developmental activities in plants [74]. When plants are subjected to salty environments, ABA plays a critical role in reducing lateral root growth while promoting primary root growth. ABA is also known to play a role in several other physiological processes, including seed dormancy maintenance and delayed seed germination regulation, seed development, stomatal closure, morphologically distinct phases in embryo development, storage protein and lipid production, leaf aging, and defense against the invading phytopathogens [75]. When root cells are exposed to a saline environment, they experience osmotic stress, which causes a rise in ABA concentration in both root and leaf tissues within a minute [76,77]. ABA promotes the synthesis of H_2_O_2_ and the expression of enzyme catalase (CAT) isoform CAT1 in response to stress. In response to H_2_O_2_, Arabidopsis MAP2K mediates this expression [78]. One of the key signaling molecules involved in stomata closure has been discovered to be ABA [79]. During this process, ABA binds to pyrabactin resistance 1/ PYR1-like/regulatory components of ABA receptors (PYR1/PYL/RCAR). After ABA binds to its receptors, the receptors have been found to engage with PP2C phosphatases and suppress their activity, allowing SnRK2s to be released from their suppression fully [80,81], which results in activating several anion efflux channels, lowering turgor pressure and triggering stomatal closure [82,83].

#### 3.5.2. Effect on Jasmonic Acid (JA) Production

Jasmonic acid (JA) is an endogenous growth-regulating chemical discovered in higher plants as a stress-related hormone [84]. Jasmonates (JAs) are fatty acid derivatives that include key molecules including JA, methyl jasmonate (MeJA), and jasmonate isoleucine conjugate (JA-Ile) [85]. Endogenous JA improves the tomato plant’s ability to endure salinity stress, mostly through maintaining homeostasis and balance among reactive oxygen species (ROS) [86]. Meanwhile, exogenous JA also increased salinity stress resistance in two maize genotypes by increasing Na^+^ elimination in root cells by lowering Na^+^ influx. The JA levels were shown to rise during the early stages of salt stress, suggesting that it could be indirectly involved in the leaf growth inhibition of salt-sensitive genotypes [87]. However, other studies have shown that exogenous JA improved photosynthetic rates, proline content, ABA levels [88], and antioxidant enzyme activity [89], or lowered Na^+^ build-up rates in shoots, thereby reducing salt-induced injury in plants [50].

#### 3.5.3. Effect on Gibberellic Acid (GA) Production

Gibberellic acid (GA) is a tetracyclic di-terpenoid molecule that stimulates plant growth and development. In addition to stimulating seed germination, GAs trigger transition from juvenile to adult leaf stage, meristem to shoot growth, vegetative to flowering, and determine sex expression and grain development [90]. Salinity decreased the endogenous bioactive GA in some plants (such as Arabidopsis seedlings), which was linked to DELLA storage. The quadruple-DELLA mutants displayed less salt-stress-induced inhibitory impact and delayed flowering, but greater salinity-mediated death, suggesting that DELLA proteins help plants survive by reducing growth in high salinity environments [91]. When seeds resume their development during seed germination, GA increases amino acid levels in the embryonic state and encourages hydrolytic enzyme production, which is necessary for the breakdown of starch present in the endosperm into monomeric units, resulting in increased plant growth and productivity [92]. In several plants, GA has been demonstrated to reduce the negative effects of salinity stress on plant development, seed germination, seed yield, yield quality, and antioxidative enzyme activity [93].

#### 3.5.4. Effect on Ethylene Production

Ethylene regulates salt stress responses by activating antioxidant defense and boosting nitrate and sulphate assimilation, in addition to controlling the equilibrium of Na^+^/K^+^, minerals, and ROS [94]. Cao et al. [95] have reported that the usage of the ethylene precursor ACC improves tolerance to salt stress, but mutations in ethylene signaling-associated genes such as *ETR1, EIN2, EIN3,* and *EIN4* produce hypersensitivity to increased salinity stress. The principal mechanism by which ethylene improves saltstress tolerance is by regulating and maintaining the ROS-generating and ROS-scavenging machinery [96].

#### 3.5.5. Effect on Salicylic acid Production

Exogenous salicylic acid has been shown to improve the ability of plants to tolerate salinity through a variety of mechanisms. These mechanisms include better photosynthetic activity [97], increased safety against ROS and other oxidants which damage the plant cells [98], inhibition of Na^+^ and Cl^–^ ions accumulation in the cells under high saline conditions [99], improvement in the assimilation of elements such as N and S, as these elements are essential for plant growth and development [100], the accumulation of soluble carbohydrates [101], stimulation of ABA aggregation which initiates signaling for stress tolerance [102], counteracting auxin signaling [103], and upgrading mineral nutrient uptake [104]. The ability of salicylic acid to cause the deposition of osmoprotectants, which mostly include amino acids such as proline, glycine betaine, and polyamines, is crucial for salt stress tolerance [105].

### 3.6. Effect on Gene Expression

Meng et al. [37] have analyzed the variations in protein expression in the chloroplasts of diploid and tetraploid *R. pseudoacacia* leaves under salt stress and reported that the proteins were differentially expressed and divided them into seven groups based on their function. Ribulose1,5-bisphosphate carboxylase/oxygenase activase, one of the proteins associated with the Calvin–Benson–Bassham cycle, was downregulated in tetraploid plants exposed to salinity stress. ATP synthase CF1 beta subunit, ATP synthase beta subunit, and chlorophyll a/b-binding 3C-like protein were among the proteins involved in photosynthetic electron transfer that showed a dramatic decrease in both, when exposed to salinity stress. In diploid plants, the above-mentioned ATP synthase beta subunit and chlorophyll a/b-binding 3C-like protein showed similar downregulation. Several chaperone proteins were also downregulated, including chaperone protein ClpC, heat shock protein (putative), and chlorophyllase-2 (putative). In diploid plants exposed to 250 mM NaCl, the expression of regulation/defense linked proteins such as lectin and ferritin-3 was shown to be increased. Apart from these, transketolase, enolase, glutamine synthetase, malate dehydrogenase, and proteasome subunit alpha type protein expression was also shown to be upregulated in both diploid and tetraploid plants, while upregulation was found to be greater in tetraploid plants compared to diploid plants. However, other proteins such as short chain alcohol dehydrogenase and tasselseed2-like short-chain dehydrogenase were downregulated in both diploid and tetraploid plants. Inconsistencies in gene transcription and protein expression levels were also discovered, indicating post-transcriptional regulation or post-translational processing [37].

### 3.7. Biochemical and Molecular Mechanism

Salt stress invokes a broad range of plant responses distressing its morpho-physiological, biochemical, and molecular mechanisms. The stressed plants’ initial and common response is the elevation of reactive oxygen species such as peroxide (H_2_O_2_) along with reactive nitrogen species such as nitric oxide (NO), which can have both positive and negative effects during stress response [106,107]. Undue accumulation of reactive oxygen and nitrogen species in response to stress can lead to oxidative stress damaging cellular structures [107]. The colorless gaseous molecule nitric oxide (NO) is a key compound involved in the physiology and development of plants and responses to drought, salinity and cold [106,108,109,110]. Nitric oxide crosstalk with other signaling chemicals and phytohormone signaling pathways has been proven to help alleviate salt stress in recent years [111,112]. Under salt conditions, sulfur and NO interact to regulate ABA and ET (ethylene) levels in the guard cell, they are also involved in the regulation of photosynthetic and stomatal activities [113]. During stress, the vital cellular functions are accomplished with a simultaneous increase of H_2_O_2_ and NO production [106,114,115]. The quinoa seed treatment with CaCl_2_, H_2_O_2_ and sodium nitroprusside at 0.2 mM, resulted in improved germination and germination index with significant mean germination time in both optimal and stress conditions along with a reduction in the negative effect of salt stress on α-amylase and β-amylase resulting in starch breakdown and increased content of water soluble sugars in salt stressed seeds [116].

The reactive oxygen species (H_2_O_2_ and/or hydroxyl radical) acts as a signaling molecule at a minimal amount [117]. Conversely, excess ROS has deleterious effects on growth and yield by causing the photo inhibition of photosystem II, photosynthetic pigment degradation, lipid molecule oxidation and inhibiting gene expression [54,55,118,119,120,121]. Plants scavenge the adverse ROS with efficient non-enzymatic and enzymatic antioxidant detoxification mechanisms. Tocopherols, ascorbate, phenolics, and glutathione are non-enzymatic antioxidants and superoxide dismutase (SOD), peroxidase (POX), catalase (CAT) and enzymes of the ascorbate–glutathione cycles which detoxify ROS [122,123,124,125,126]. The ability of plants to survive in the stressful environment depends on the signaling networks and their crosstalk [127] with activities activated after sensing of a signal by the specified receptor which triggers the influence of further signals and protein phosphorylation cascades, such as MAPK signaling [128]. The function of MAPK relies on post-translational phosphorylation signaling, established by a serine/threonine kinase, i.e., mitogen-activated protein kinase kinase kinase (MAPKKK or MAP3K) that reversibly phosphorylates MAPKK (a dual-specificity kinase), that then phosphorylates MAPKs [127]. At this juncture, signaling compound crosstalk, including gasotransmitters such as nitric oxide, hydrogen sulfide, hydrogen peroxide, calcium, reactive oxygen species along with growth regulators, auxin, ethylene, abscisic acid and salicylic acid has a key role in regulating stress signaling and govern unfavorable situations, such as salt stress. Recent developments in multiomics technology, transcriptomics, genomics, proteomics, and metabolomics have facilitated the highlighting of an in-depth perspective in multiple stress tolerance [129].

Plants possess a large number of genes and are acutely complex in how they react to salinity, thus it is difficult to study completely how plants react to salinity owing to their multi-genetic nature [130]. Three novel QTLs were identified on chromosomes 4, 6, and 7 associated with salt tolerance in rice through a genome-wide association study [131]. Advances such as zinc finger nuclease, transcription activator-like effector nucleases, CRISPR-Cas9 [132] and speed breeding [133] hold promise in developing engineered crops for salt tolerance. Genes related to stress tolerance can be isolated by analyzing the difference in transcriptional levels among tolerant and sensitive genotypes under stress. Various up and downregulated transcriptional factors such as MYB, MYB-related, AP2-EREBP, NAC, and WRKY were revealed by transcriptional profiling to study a key component in the salt tolerance network in developing salt tolerant plants [134]. In recent trend and future perspectives, it is believed that genetic resources and integrated ‘omics-assisted’ approaches such as phenomics, ionomics, transcriptomics, proteomics, genomics, miRNAomics, lipidomics, and metabolomics would be significantly employed for developing salt tolerance in crop species globally [135], along with epigenetics and next-generation phenotyping [136].

### 3.8. Eco-Physiological Aspects and Salinity Stress

Among many other stress factors, the key problem of worldwide soil salinization is increasing every day due to a climate change-induced rise in sea levels, intensive irrigation techniques using saline water, and large-scale soil erosion [137]. Around 20% of farmland is saline, and as a result of the major issue of global warming, even more agricultural landis becoming saline [12]. The idea of sustainable intensification of agriculture (SIA), which involves increasing agricultural output while minimizing environmental impact, is an important factor to consider in this regard [138,139]. Climate change impacts the intensity of various stresses, such as salinity and their resultant effects on plant growth and crop yield. Salt stress is exacerbated by three majorfactors: (i) global warming, (ii) inefficient agricultural field drainage, and (iii) increasing water table [23]. Increased air temperature shortens the growing season, leading to lower yields. As a result, the impact of climate change on yield decrease seems to be more significant in locations where temperatures are higher [140]. Low rainfall can also influence the severity of salinity due to decreased soil leaching and increased evapotranspiration, which enhancessoil salinity [141].

## 4. Salt Overly Sensitive (SOS) Signaling Pathway

Plants have developed a variety of physiological and biochemical processes to help them to thrive in high-saline environments. Because soil contains salts, the main focus of research should be on the transport mechanism of the Na^+^ ion and its compartmentalization [142]. Ion homeostasis is defined as the transport and balancing of cytosolic ion concentration by the plasma membrane and its channel proteins, antiporters, and symporters. The Na^+^ enters the plant passively through the root endodermis or different channels under saline conditions [12]. Plants can adjust in two ways to avoid high levels of Na^+^ in the cytoplasm of root cells: (1) Na^+^ exclusion from root uptake and (2) Na^+^ sequestration in vacuoles [73]. Exclusion, redistribution, elimination, succulence, and accumulation in the cytoplasm maintain Na^+^ concentration in plants until their osmotic potential is lower than that of the soil. Holding back the excess accumulation of Na^+^/K^+^ and regulating the water flux are critical steps in initiating ion homeostasis [143].

Many researchers have focused their efforts on various pathways that are activated after a plant is exposed to high salt concentrations, and it has been discovered that the salt overly sensitive (SOS) stress signaling pathway is activated in order to maintain the ionic concentration and eventually achieve salt tolerance [144] and SOS1, SOS2, and SOS3 are the three key proteins in the SOS pathway (Figure 2). Many transporters maintain Na^+^ homeostasis in root cells by preventing excess Na^+^ ions from entering the cell cytoplasm, with the SOS1 antiporter being one of the most significant [145]. SOS1 is a plasma membrane Na^+^/H^+^ antiporter/exchanger that removes excess Na^+^ from root cells in the rhizosphere, which is why it is necessary to reduce ionic stress. The cytosolic C-terminal tail of the SOS1 protein is 700 amino acids long and it contains a nucleotide-binding motif as well as an auto-inhibitory domain region containing a serine residue that is phosphorylated by SOS2 [146].

The SOS1 is auto-inhibited in normal conditions when there are no hypersaline conditions. This auto-inhibition is only relieved when the SOS2 protein adds a phosphate group to the Ser1044 residue in the C-terminal region of SOS1, which occurs when the plant is under salinity stress [146]. The SOS2 gene is responsible for producing a serine/threonine kinase enzyme [147]. The FISL/NAF motif, which comprises 21 amino acids, is found in the regulatory region of SOS2 at the C-terminus, while the catalytic domain is found at the N-terminus, showing sequence homology with sucrose nonfermenting (SNF) kinases [148]. The interaction of the regulatory domain containing the FISL motif with the catalytic domain of SOS2 causes auto-inhibition in normal conditions. However, the calcium-dependent SOS3 protein, which interacts with SOS2 through its regulatory domain, activates SOS2 during stress [149].

Salt stress induces an increase in cytosolic Ca^2+^ levels, and SOS3 can detect this change in calcium level, causing SOS2 to bind to SOS3, resulting in the formation of SOS2-SOS3 complex [148]. The SOS3, a Ca^2+^ binding protein with a myristoylation site at its N-terminus, is the third protein in this pathway [144]. As a result of its interaction with the regulatory domain of SOS2, the SOS3 plays a key role in activating and recruiting SOS2 to the plasma membrane. On SOS3, a seven amino acid region called MGXXXS/T(K) undergoes N-myristoylation and is essential for loading the SOS3-SOS2 complex onto the plasma membrane. The SOS3-SOS2 complex activates the SOS1 via the myristoylated N-terminus motif of SOS3 [150]. The SOS3-like Ca^2+^ binding protein 8 (SCaBP8) is another protein with the same function as SOS3, but with a different site of action: SCaBP8 is mostly active in the shoot, whereas SOS3 is primarily active in the root. After sensing altered calcium levels, both SOS3 and SCaBP8 proteins activate SOS2 via the FISL motif, which results in recruitment of the resultant complex to the plasma membrane and hence activation of SOS1. The major difference between the two proteins is that SCaBP8 lacks the seven amino acid sequence where myristoylation occurs, whereas SOS3 has it [151].

The SOS1 is responsible for transporting Na^+^ ions from epidermal cells in roots to cells in the xylem parenchyma and ultimately to leaves, where they are sequestered in vacuoles. However, the meristematic root tip cells lack vacuoles and have SOS1 in their epidermis, which allows them to excrete Na^+^ directly into the soil [152]. Through the action of vacuolar ATPases, SOS2 also accumulates excess Na^+^ ions in the vacuoles. This is accomplished by binding to regulatory units that control the Na^+^/H^+^ exchange. There are two types of antiporters in the tonoplast (plasma membrane of vacuoles). Vacuolar-type H^+^-ATPase (V-ATPase) and vacuolar pyrophosphatase (V-PPase) are the two enzymes involved. In *Arabidopsis*, the SOS stress signaling pathway has been widely researched, and the proteins involved in this process have been identified as SOS3/SCaBP8–SOS2–SOS1 signaling to regulate Na^+^ exclusion and cellular ion homeostasis [73].

## 5. Ethylene

Ethylene is a gaseous signaling molecule that regulates stress responses as well as other developmental processes (fruit ripening, abscission of petals and leaves, senescence of flowers, stimulation of root formation, and inhibition of seedling elongation) in plants [153,154]. Various hormones have been found to influence plant growth, but ethylene was the first gaseous hormone found in plants to play a critical role in regulating plant growth and development under various stress conditions, including salinity stress [94,155]. Because it is gaseous, it can easily permeate into neighboring cells, even though its production takes place at the site where it must execute its hormonal function. Ethylene production induces three critical responses, collectively known as the triple response, including inhibition of hypocotyl and root extension, hypocotyl swelling, and enhanced apical hook tightness [156]. The closing of stomata influences the plant abiotic stress response, and this stomatal closure is controlled by a complicated signaling system that leads to enhanced stress tolerance [157]. Although ABA is recognized as a critical regulator of stomatal closure under abiotic stress, multiple investigations have found that ABA and ethylene have antagonistic roles in stomatal movement control [158,159].

### 5.1. Ethylene Biosynthesis

There are only two processes in the production of ethylene (Figure 3). The biosynthesis of ethylene begins with the formation of S-adenosyl-L-methionine (AdoMet) from the precursor methionine, which is catalyzed by the enzyme S-adenosyl-L-methionine synthase. The rate-limiting step is the conversion of AdoMet to ACC (ethylene precursor) by the enzyme ACC synthase (ACS) [160]. The production of ethylene from ACC is an exothermic reaction in which energy is released as heat through oxygen. Because the ACS enzyme catalyzes the rate-limiting step in ethylene production, it is regulated. This enzyme is post-translationally regulated; it is first phosphorylated and subsequently destroyed by ETO1 and CUL3, which add ubiquitin molecules to the protein, which is then targeted for destruction [161].

The salt increases ethylene synthesis in various plant species by altering the activity of ACS and ACC oxidases (*ACO*) [162,163]. After analyzing several stress-responsive genes, it was observed that this change could be linked to epigenetic changes that affect the expression of *ACS* genes [164]. The Arabidopsis genome comprises 12 putative ACS-like genes; one of these *ACS* genes, *ACS3*, was shown to be a pseudogene and was identified as such by a short sequence; also, *ACS12* and *ACS10* are known to code for an aminotransferase enzyme without the catalytic activity of *ACS* [165]. The remaining nine ACS-like genes coding for ACS proteins can be classified into three kinds based on the presence or lack of phosphorylation sites at the C-terminal [166].

Type-1 ACS proteins have a long C-terminal domain with shared action sites and conserved sequences for two distinct kinases, mitogen-activated protein kinase (MAPK) and calcium-dependent protein kinase (CDPK). The TOE (target of ETO1, the ethylene overproducer 1 protein) regulatory motif overlaps the CDPK target site in the second set of proteins, type-2 ACS proteins. Type-3 ACS proteins have only a brief sequence of amino acids in their C-terminal domain and no MAPK or CDPK target sites [167]. There is a gene family known as the ethylene response factors (ERFs) among the environmental stress-responsive genes. The mRNA levels of several ERFs are known to be regulated by several compounds produced and hormones acting in diverse stress conditions [168]. Because of the stimulation and activation of *ACS*, several environmental stresses, including salinity, cause ethylene synthesis and aggregation. As a result, PIF4 regulates the transcriptional activation of *ACS* genes in response to abiotic stressors, including salinity and drought [169]. The transporter LHT1 absorbs the soluble ACC and distributes it throughout the plant via the xylem [170,171].

### 5.2. Ethylene Signaling

Receptors detect ethylene on the membrane of the endoplasmic reticulum. It then proceeds through a signaling pathway involving signal transduction into the nucleus, resulting in changes in the expression of genes impacted by ethylene [172]. *Ethylene response 1 (ETR1), ETR2, ethylene response sensor 1 (ERS1), ERS2,* and *ethylene insensitive 4 (EIN4)* are the five ethylene receptors studied so far in Arabidopsis. These receptors negatively regulate the ethylene signaling in various plants, and they have only been found in the presence of salinity stress [172]. At their N-terminus, all receptors feature three to four membrane-spanning helices located on the plasma membrane of the endoplasmic reticulum and these helices include the ethylene binding site [173].

When ethylene isnot present, the receptors become active, and the function of these receptors can be controlled by a complex made up of two proteins: reversion to *ethylene sensitivity 1 (RTE1)* and *auxin-regulated gene involved in organ size (ARGOS)*. They act as a negative regulator of ethylene sensitivity by positively regulating ethylene receptors [174,175]. In the absence of ethylene, active receptors have been demonstrated to bind to *constitutive triple response 1 (CTR1)* protein, resulting in its activation (Figure 4) [176]. A slightly higher ethylene concentration increases receptor gene transcription and stabilizes *CTR1*, whereas a higher ethylene concentration causes receptor/*CTR1* to be destroyed by proteasome-mediated degradation [177]. When coupled to ethylene receptors, *CTR1* is a serine-threonine kinase that remains activated. *CTR1* phosphorylates and consequently deactivates *EIN2*, an endoplasmic reticulum membrane protein, in its active form. Two F-box proteins, such as *ethylene insensitive 2 targeting protein 1 (ETP1)* and *ETP2*, bind to this phosphorylated and deactivated *EIN2*, causing 26S proteasomal degradation of the target protein [178]. The *EIN* proteins are transcription factors that bind to a certain gene and change its expression, and they also carry on the signals in the ethylene signaling pathway downstream of the *CTR1* protein [179].

In addition to the five ethylene receptors, salinity stress is known to influence CTR1. Studies have shown that mutants with CTR1 loss-of-function can resist saline conditions related to manipulating the shoot Na^+^/K^+^ ratio, which is primarily regulated by ETR1-CTR1 signaling [180]. It was noted that copper (Cu^2+^) ions are required for ethylene to bind to its receptors under stress conditions and this requirement for Cu^2+^ ions is fulfilled by a transporter called *responsive to antagonist 1 (RAN1)*, which aids in the transfer of Cu^2+^ ions by ethylene. When ethylene is present in significant amounts, it binds to its receptors, causing *CTR1* to be inactivated, *EIN2* to be dephosphorylated, and *EIN3/EILs* to be activated in enhanced ethylene responses [181]. When ethylene is present, *EIN3* and *EIL1* boost the expression of ERFs and transcription factors (Figure 5) [182]. The salinity stress-induced stability of *EIN3/EIL1* improves salinity tolerance by reducing ROS accumulation in plants [96].

It is well known that high-salt soil has a detrimental impact on seed germination, which negatively impacts plant growth and net yield [12]. Different components and factors of ethylene signaling have a beneficial or detrimental impact on the seed germination process and subsequent growth in a saline environment [183]. Seed germination was found to be negatively influenced by ETR1 and EIN4 in Arabidopsis plants. However, ETR2 favorably regulated the process and was found to enhance seed germination under salinity stress [184]. Elongated hypocotyl 5 (HY5) protein upregulates *ABA insensitive 5 (ABI5)* gene expression and this increase of *ABI5* is responsible for limiting and preventing seed germination. *EIN3/EIL* degrades the HY5 protein, allowing constitutive photomorphogenesis 1 (COP1) to reach the nucleus [185]. During salinity stress, after HY5 is broken down, ABI5 expression is halted, and seed germination inhibition is consequently stopped [94].

Ethylene-responsive element binding factors (ERFs) are transcription factors that play a role in the ethylene signaling cascade downstream of *EIN3* [94]. *ERF1, ERF2, ERF5, ERF6, ERF8, ERF9, ERF11, ERF59, ERF98,* and *RAP2.6L* have been shown to play a role in the transcriptional cascade that governs the suppression of leaf growth when exposed to mild osmotic stress [186]. The majority of ERFs are positively influenced by activating the response, and their concentration rises quickly after exposure to stress, although two ERFs, *ERF8* and *ERF9*, are negatively influenced. After some time, these two ERFs are engaged to reduce the risk of over activation and allow the fine-tuning of the stress response. This signaling is transcriptionally activated in response to osmotic stress, but it has also been reported to work in response to abiotic stress factors such as salinity and drought stress. *ERF8* has a substantial inhibitory effect on leaf cell proliferation and growth [187]. It has been observed that after being phosphorylated via the MPK3/6-cascade to regulate ethylene production, the activity of some ERFs increases, resulting in dual level response regulation mediated by ERFs [188].

## 6. Role of ACC Deaminase to Overcome the Salinity Stress

The presence of ACC deaminase, encoded by the *acdS* gene and having 325–345 amino acid residues, has been found in all three domains of life: archaea, bacteria, and eukarya. It is a multimeric enzyme of monomeric subunits with a molecular weight of 33–42 kDa that can function as a homodimer, homotrimer, or homotetramer [189,190].

### 6.1. Structure

X-ray crystallographic analysis revealed that ACC deaminase folds into two domains, each with an open twisted α/β structure similar to the β-subunit of tryptophan synthase [191]. When structural analysis using nuclear magnetic resonance, X-ray crystallography [192] and mutagenesis was carried out in yeast [193] and *Pseudomonas* [194,195], the residues mainly constituted the enzyme’s active site viz., Tyr269, Tyr295, Lys51 and Glu296 (yeast ACC deaminase numbering sequence), while similar residues viz., Tyr268, Tyr294, Lys51 and Glu295 were found in *Pseudomonas* sp. UW4. The amino acid residues that make up the enzyme’s active site have important functions. For example, Lys51 is involved in ACC proton extraction, while Tyr294 is a catalytic residue that helps position Pyridoxal 5′-phosphate (PLP) cofactor correctly within the active site and aids in external aldimine formation by interacting with the amino group of the substrate [192,194].

### 6.2. Enzyme Biochemistry and Its Function

ACC deaminase is a pyridoxal 5-phosphate-dependent tryptophan synthase beta super family (fold type II) hydrolase that is involved in the breakdown of ACC, an immediate precursor associated with ethylene synthesis via the methionine pathway, by employing the cofactor pyridoxal phosphate [190,192]. Because the enzyme’s activity relies on its substrate, ACC, the enzyme is inducible. ACC deaminase activity can be induced by ACC at levels as low as 100 nM in *Pseudomonas* sp. with complete induction up to 10 h [196]. The enzyme degrades ACC by opening the cyclopropane ring and deaminating it, resulting in the production of α-ketobutyrate and ammonia [17,197,198,199]. As a result, when plants are inoculated with PGPR, which produces this ACC deaminase, the inhibitory ethylene levels in plants under stress are reduced.

### 6.3. Mechanism of Action of the Enzyme on Its Substrate ACC

Two reaction mechanisms have been proposed: direct β-hydrogen abstraction is the first mechanism and indirect β-hydrogen abstraction by nucleophilic addition is the second mechanism. Internal aldimine is formed when the Lys residue of the ACC deaminase enzyme combines with the PLP cofactor in both cases. Transaldimination occurs next, in which internal aldimine is acted on by ACC and transformed to external aldimine via an aminyl intermediate. Both reaction mechanisms differ in terms of the reactions that lead to quinoid formation, which can be either direct or indirect β-hydrogen abstraction, resulting in the formation of products such as aminocrotonate and quinoid, which reversibly hydrolyzes to generate α-ketobutyrate and ammonium, culminating in internal aldimine regeneration [197,200].

The methylene proton undergoes direct β-hydrogen abstraction via the Lys51 residue of ACC deaminase, forming a quinonoid, where electronic rearrangements and protonation occur, forming another quinonoid, which is nucleophilically attacked by the protein’s basic residue. However, during indirect β-hydrogen abstraction, the formation of external aldimine is followed by a nucleophilic attack through a basic residue of the pro-S β-carbon of ACC, as well as the removal of a proton from another nearby basic residue, resulting in ring opening and the formation of a quinoid [197,200]. Because of its electrical structure and thermodynamic favorability, the indirect mechanism is more favorable as a primary reaction used by ACC deaminase. To elucidate this, a mutant enzyme was utilized to isolate the reaction intermediates because the direct mechanism does not adequately account for the mechanism of action of the enzyme in its original form. Furthermore, the reaction is predicated on an inert proton abstraction followed by anion-induced cleavage, which is stereoelectronically unfavorable [200].

### 6.4. Transcriptional Regulation of ACC Deaminase Gene (acdS)

The expression of the *ACC deaminase gene* (*acdS*) is influenced by the type of organism and the surrounding environment. At least one of these factors, namely (i) leucine-responsive regulatory protein (LRP) coupled with cyclic AMP receptor protein (CRP) and fumarate-nitrate reduction regulatory protein (FNR), (ii) *nitrogen fixation (nifA)* genes, (iii) *RNA polymerase sigma S (rpoS)* gene and (iv) other modes of enzyme regulation, regulates the transcription of these *acdS* genes [190]. Other factors, including oxygen availability, substrate concentration, and product concentration affect the expression of this enzyme in addition to these transcriptional factors. Various investigations have shown that different phylogenetic groupings use different mechanisms to regulate this gene’s expression, implying that these modes of regulation are quite complex [190,191].

There are several ways for transcriptional regulation and expression of the *acdS* gene in bacteria, as explained below.

#### 6.4.1. LRP Coupled with CRP and FNR

The acdS gene is regulated by several transcription factors working together. The CRP box, FNR box, *acdB* gene, *acdR* binding site, and *acdR* gene expressing open reading frame (ORF) were among the regulatory elements discovered in an extensively investigated *Pseudomonas* sp. UW4 strain [201,202,203,204]. The *acdR* gene (ACC deaminase regulatory gene encoding LRP protein) transcription and expression are promoted in the presence of ACC, resulting in the synthesis of LRP [190]. In its active state, LRP binds to a complex formed by ACC and glycerophosphoryl diester phosphodiesterase, both expressed by the *acdB* gene, to form a tripartite regulatory complex [190,204]. This tripartite regulatory complex then binds to either FNR box (if O_2_ levels are low) or CRP box (if O_2_ levels are high) and activates the *acdS* promoter region (P2 or P3), causing the initiation of *acdS* gene expression [190].

In bacteria that lack CRP or FNR box, the regulatory complex binds ACC directly to the *acdS* promoter [205]. As a result, ACC deaminase is produced, breaking down the substrate ACC into ammonia and α-ketobutyrate, forming branched-chain amino acids such as leucine. When the concentration of leucine rises, it attaches to the active LRP octamer, transforming it into an inactive dimer-leucine complex, which helps suppress the expression of the *acdS* gene. The *acdS* gene is regulated so that it only transcribes when needed (Figure 6) [204].

#### 6.4.2. Nitrogen Fixation (nifA) Genes

When the bacterial strains produce the ACC deaminase enzyme but the DNA sequence does not encode for an *acdR* gene, the transcription of the *acdS* gene is controlled by the *nifA* promoter, which is associated with transcriptional regulation of *nitrogen fixation (nif)* genes, as in the case of a few strains of *Rhizobia* and *Mesorhizobium* (Figure 7) [190,206]. The components present are the regulatory N_2_ fixing units, namely *nifA1* and *nifA2*, which are present upstream of *acdS* and *nifH* in *Mesorhizobium loti*. The *nifA2* promoter, which codes for the *nifA2* protein, interacts with the σ54 RNA polymerase, permitting *acdS* transcription [197]. The *nifA1* gene has also been identified to regulate *acdS* gene expression; however, its exact role is unknown [206]. This mechanism aids nodules containing such bacterial strains in controlling high ethylene levels, preventing senescence in the plant at an early stage [190].

#### 6.4.3. RNA Polymerase Sigma S (rpoS) Gene

The expression of many genes produced when bacteria are in the stationary phase of their growth or response to various stress stimuli is largely regulated by the sigma factor *rpoS*, an important stress modulator in β and γ Proteobacteria [207,208,209]. In a study to better understand the relationship between *rpoS* and *acdS* gene expression, researchers found that over-expression of the *rpoS* gene resulted in a 30% increase in ACC deaminase levels in a genetically modified ACC deaminase-positive strain of *Enterobacter cloacae* CAL2 with numerous *rpoS* gene copies on a plasmid [207]. The contrasting result was also observed when the same experiment was performed on *Pseudomonas* sp. UW4, the levels of ACC deaminase were found to be 20% lower when compared to the untransformed wild type [207]. Although the genes are 96% similar, the *acdS* gene and *rpoS* gene have a positive and negative correlation in both cases, demonstrating how separate transcriptional regulators control the process [210,211].

#### 6.4.4. Other Modes of Regulation

Despite the complete absence or partial expression of the *acdR* gene, some ACC deaminase-positive bacteria produce the enzyme [190]. Even though they are almost 9 kb away from the *acdS* gene, *acdR* or a similar gene that produces LRP that can influence the expression of *acdS* gene [191]. Different species of bacteria, even though they belong to the same genus, *Burkholderia*, differ in terms of transcriptional regulation, with the two strains *Burkholderia* sp. CCGE1002 and *Burkholderia phymatum* STM815 lacking the *acdR* gene but having two *acdS* gene copies, one on the bacterial chromosome and the other on the mega plasmid [197]. In *B. xenovorans* LB4000, another species in the same genus, the transcription of *acdS* is controlled by the LysR family of regulatory elements, which is a fundamentally distinct mechanism of regulation [141]. In addition, *Brenneria* sp. EniD312, *Dickeya* sp., and *Pantoea* sp. At-9B is an example of *Acinetobacter* sp. and *Proteobacteria* sp. that use the LysR family of regulatory elements to control the transcription of this gene [191]. The *acdS* gene is regulated by a gene related to the regulatory protein GntR in other *Actinobacteria* and *Meiothermus* species. Other mechanisms involving operon regulatory factors like M20 peptidase may have regulated this gene in *Saccharopolyspora erythraea* NRRL 233 and *Streptomyces hygroscopicus* ATCC 53653 *acdS* is part of the major facilitator super family (MFS) proteins, which are also controlled by the same operon regulatory factors [191].

## 7. Plant Growth-Promoting Rhizobacteria (PGPR)

At each stage of its growth and development, a plant might be harmed by biotic and abiotic stressors. Though biotic stresses can be avoided or treated with chemical components, abiotic stresses are inescapable and difficult to manage, resulting in a loss of more than 50% for most crop plants [212,213]. PGPR has been a particularly effective approach for reducing the bad effects generated by both types of stress factors, among the various methods designed to deal with such situations. The PGPR is a type of bacteria that lives in the rhizosphere, a specialized area around the roots of a range of leguminous and non-leguminous plants, and performs many biological and ecological functions [214]. The rhizosphere is a tiny soil zone surrounding the root system with a higher concentration of important and helpful nutrients than the rest. This is due to a large number of disseminated plant components such as amino acids and sugars. These disseminated components are high in nutrients and energy, which have a beneficial impact on various microbes’ growth and metabolic activities, resulting in the region being heavily flooded with bacteria [215].

The plant-to-microbe signal molecule, genistein, has been used for the alleviation of salt stress during (N)-fixation in soybean by a species of legume-root nodulating, micro symbiotic nitrogen-fixing bacteria *Bradyrhizobium japonicum* by inducing particular bacterial genes allowing the bacteria to progress through the N-fixation process and fix atmospheric nitrogen while being environmentally and economically viable [216,217,218] along with other soil microbes including plant growth-promoting rhizobacteria and mycorrhizal fungi under both greenhouse and field conditions [219,220]. In wheat, it has been observed that resistant species use a range of mechanisms to mitigate salt stress, including sodium exclusion, osmoregulation and potassium retention. The most crucial are biotechnology-based crop breeding, seed priming and soil microbes, using resistant genotypes or combining these methods, and the scientific use of irrigation water [140].

The PGPR is divided into extracellular PGPR (ePGPR) and intracellular PGPR (iPGPR). The ePGPR (such as *Serratia, Azospirillum, Azotobacter, Bacillus, Chromobacterium, Caulobacter, Agrobacterium, Erwinia, Pseudomonas*, etc.) reside and function outside of plant root cells, in soil intimately linked with roots (rhizosphere), on root surfaces (rhizoplane), and in spaces between root cortex cells. The endophytes/symbionts of the iPGPR (such as *Rhizobium, Bradyrhizobium, Allorhizobium, Mesorhizobium*, etc.) as endophytes/symbionts live inside the plant root cells. These PGPR are known to synthesize extracellular hydrolytic enzymes such as chitinases, glucanases, cellulases, and proteases that cause cell lysis and the destruction of fungal cell walls, one of which is ACC deaminase [214]. This unique group of rhizobacteria is also known to produce biosurfactants, which can negatively act on pathogenic microbes by disrupting the permeability of their plasma membrane, resulting in cell lysis. They also produce siderophores, which can slow the growth of pathogenic organisms by reducing iron availability [214,221].

Under high salt conditions, the enzyme ACC deaminase has been shown to play a critical role in nodule formation, promoting the persistence of infectious conditions that would otherwise be harmed by extremely raised amounts of ethylene, which aids in nodule formation [222]. Many PGPR live on the surface of roots and thrive in the areas between the root hairs and the rhizosphere’s epidermal layers. However, it has also been discovered that some PGPR species do not physically contact the roots. Chemicals secreted from roots, also known as root exudates, are the most significant aspect of rhizospheric signaling and control beneficial interactions between plants and microbes. Many secondary metabolites released by plant root cells, such as phenolic compounds, flavonoids, and other organic acids, act as chemical messengers, causing bacterial cells to migrate towards these secondary metabolites, a process known as bacterial chemotaxis. They also help with exopolysaccharide secretion, quorum sensing (bacterial cell signaling), and biofilm development during rhizosphere invasion [223].

Plant hormones, exopolysaccharides, rhizobitoxine, and lipochito-oligosaccharides are all known to be produced by PGPR [224,225]. Rhizobitoxine inhibits the production of ethylene in plants, allowing them to develop more quickly under stress. *Pseudomonas, Bacillus, Enterobacter, Agrobacterium, Streptomyces, Klebsiella,* and *Ochromobacter* are the best-known PGPR for increasing agricultural yield in saline environments [226]. The potential of halotolerant plant growth-promoting rhizobacteria (HT-PGPR) to endure and reduce salinity stress in plants is well established. Several HT-PGPR species, including *Arhrobacter, Azospirillum, Alcaligenes, Bacillus, Burkholderia, Enterobacter, Microbacterium, Klebsiella, Pseudomonas, Streptomyces, Rhizobium,* and *Pantoea,* have been observed to help crops cope with salinity stress [227]. After being inoculated with a salt-tolerant PGPR (ST-PGPR) strain of *Enterobacter* sp. UPMR18 that can synthesize ACC deaminase, the authors of [228] found that increased production of ROS as well as scavenging enzymes such as superoxide dismutase (SOD), ascorbate peroxidase (APX), and catalase (CAT) and the upregulation of ROS signaling genes led to improved crop production. The activation of the ACC deaminase gene in ST-PGPR after exposure to high saline conditions is a common phenomenon, and as a result, the ethylene concentration, which rises due to salinity stress, is reduced by ST-PGPR. This is accomplished by inhibiting the ethylene-induced downregulation of genes associated with plant stress and the upregulation of genes associated with plant growth [229].

### Role of ACC Deaminase Producing PGPR in Alleviating Salinity Stress

Ethylene is the main hormone produced when plants respond to salinity stress. When the amount of stress ethylene produced exceeds a particular threshold, it has a negative impact on plant growth. The PGPR employs various techniques to alleviate plant stress, one of which is using an enzyme called ACC deaminase, which helps lower ethylene levels. The PGPR with more than 20 nmol of α-ketobutyrate mg^−1^ h^−1^ of ACC deaminase activity promotes plant growth by significantly reducing stress ethylene in stressed plants [230]. The mechanism of action of this enzyme is to degrade ACC to α-ketobutyrate and ammonia, which are then utilized as a source of nitrogen and carbon by bacteria, allowing the plant to resume development by lowering stress ethylene levels [196,231]. Further, the application of PGPR possessing the ACC deaminase activity in inducing salt tolerance is listed in Table 1 and their beneficial role in plant growth is depicted in Figure 8.

## 8. Conclusions and Future Prospects

Soil salinity is one of the key abiotic stressors that reduce agricultural output by slowing plant growth. One reason for this could be the growing use of chemical fertilizers, which negatively modify soil composition. Plants have their mechanisms for dealing with salinity stress, but in many cases, these processes are insufficient to keep plant growth and development from being considerably hampered. Several strategies for reducing saline stress in agriculture, including chemical and physical treatment of seeds before planting and sustainable agricultural management practices, have helped reduce the effects of excessive salt accumulation in soil. Fortunately, all plants interact with microorganisms in the rhizosphere, phyllosphere, and endosphere, positively impacting plant growth. The PGPR is of special interest to crops because it includes many beneficial mechanisms, including ACC deaminase, which can boost plant tolerance to salt stress by cleaving ACC, a direct precursor of ethylene. The importance of ACC deaminase activity in bacterial strains reduces ethylene levels in plants under salinity stress has been well recognized. Undoubtedly, PGPR with ACC deaminase activity (among other mechanisms) is a viable approach for improving crop quality and yield in saline soils.

## Figures and Tables

**Figure 1 ijms-22-11461-f001:**
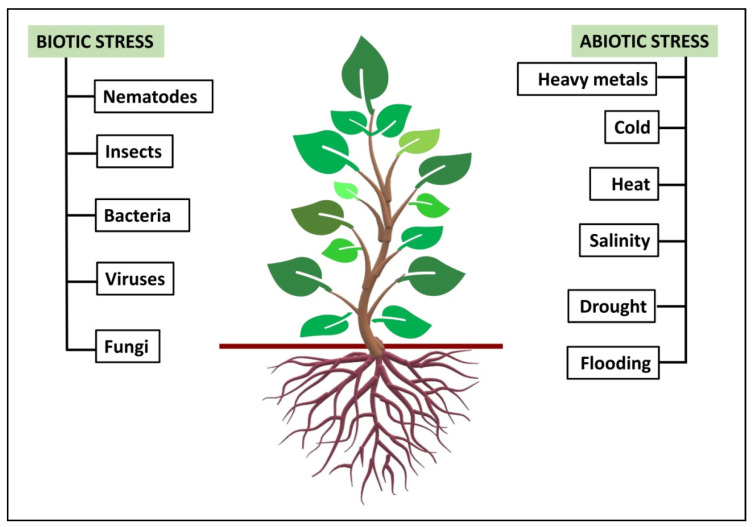
Different types of biotic and abiotic stresses that retard the growth and development of plants.

**Figure 2 ijms-22-11461-f002:**
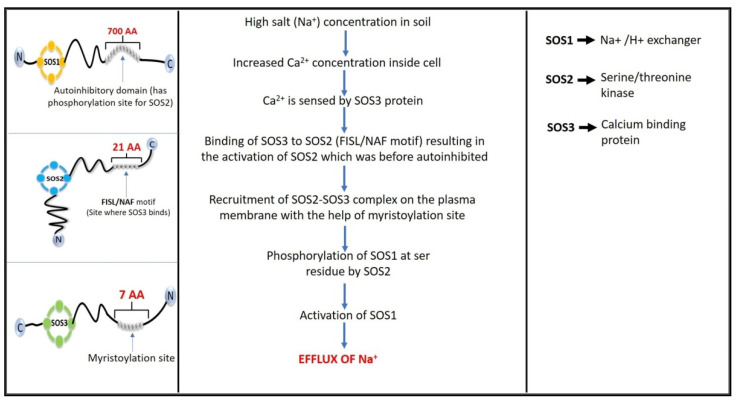
SOS proteins involved in SOS pathway which is primarily activated during salinity stress.

**Figure 3 ijms-22-11461-f003:**
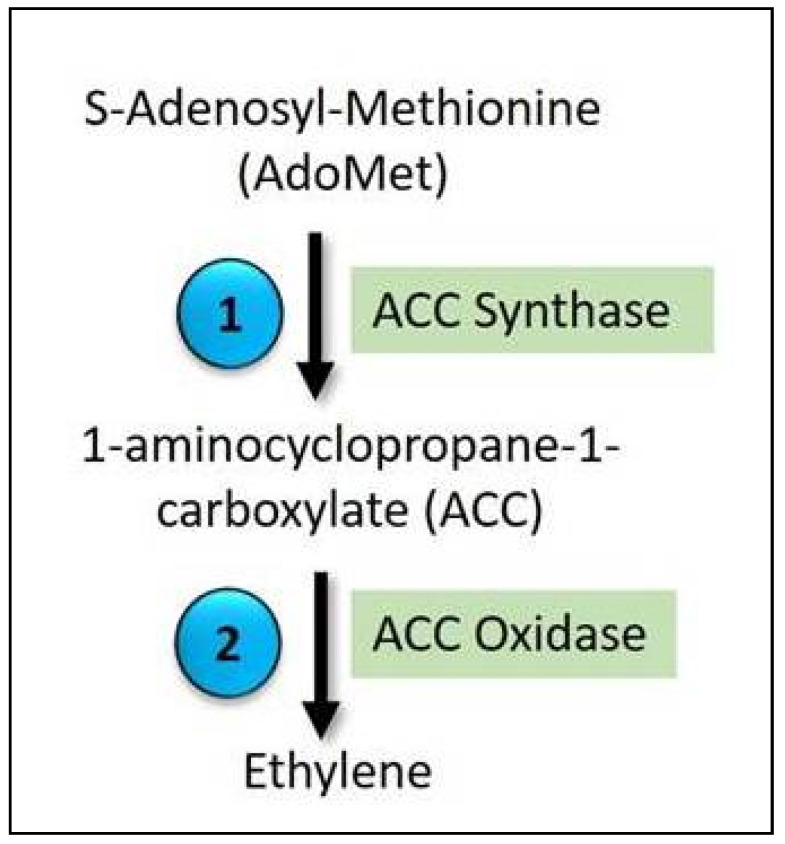
Steps involved in the biosynthesis of ethylene.

**Figure 4 ijms-22-11461-f004:**
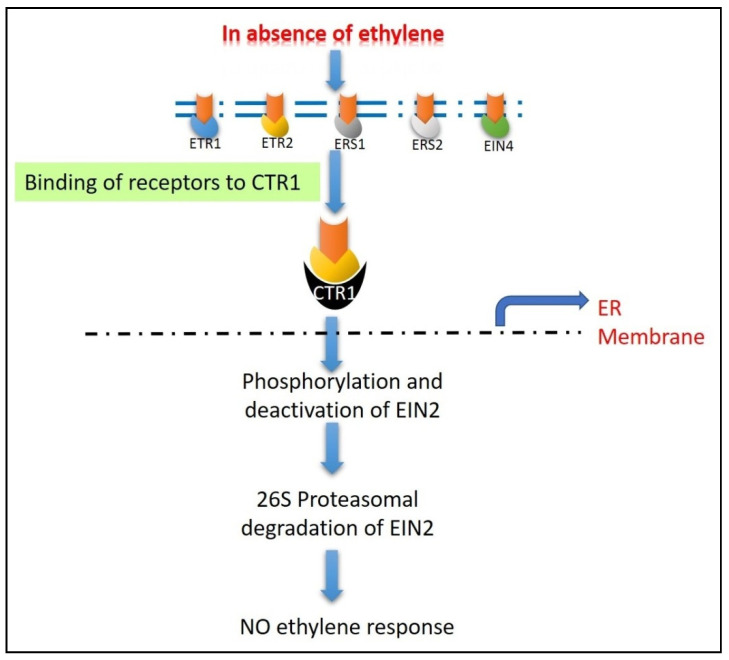
Signaling pathway involved in the absence of ethylene.

**Figure 5 ijms-22-11461-f005:**
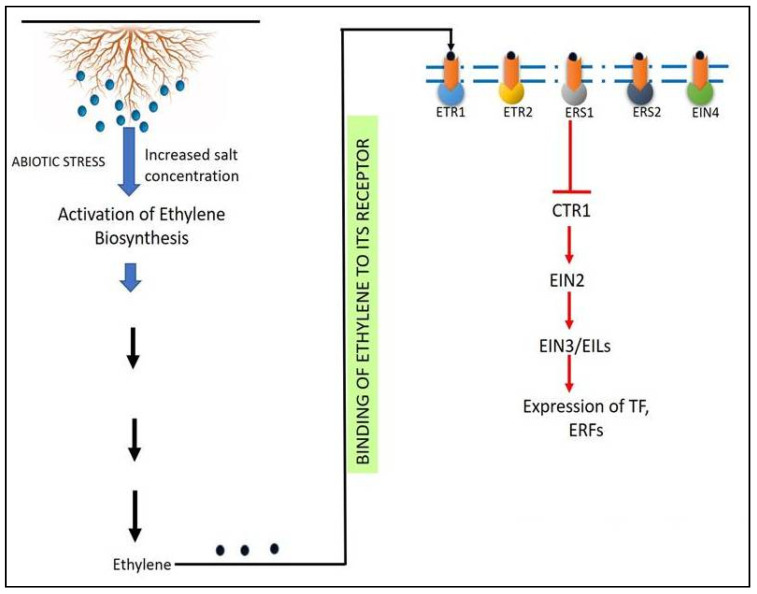
Ethylene signaling pathway under abiotic stress.

**Figure 6 ijms-22-11461-f006:**
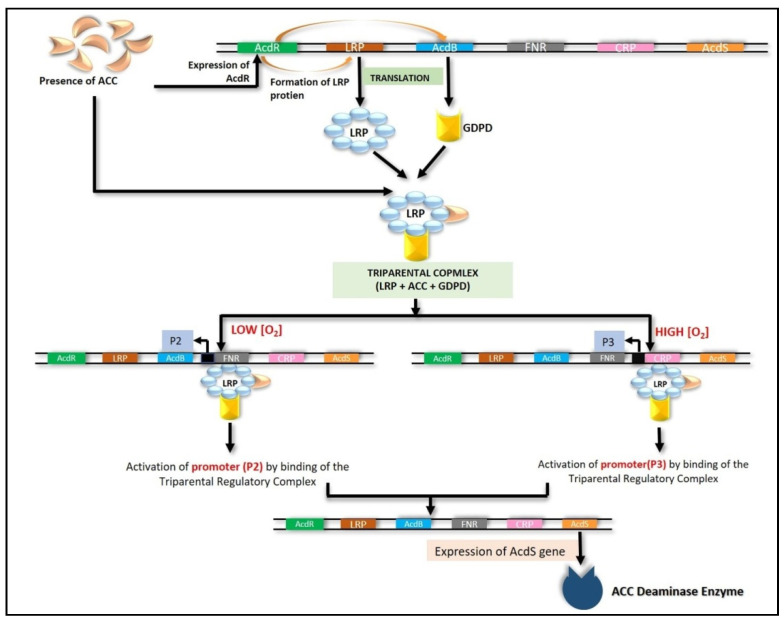
Transcriptional regulation of *acdS* gene by the action of octameric LRP protein.

**Figure 7 ijms-22-11461-f007:**
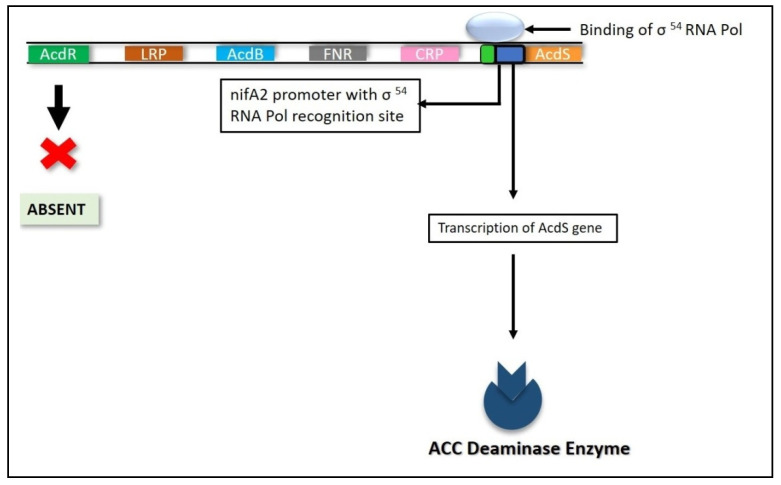
Regulation of *acdS* gene by *nifA2* promoter.

**Figure 8 ijms-22-11461-f008:**
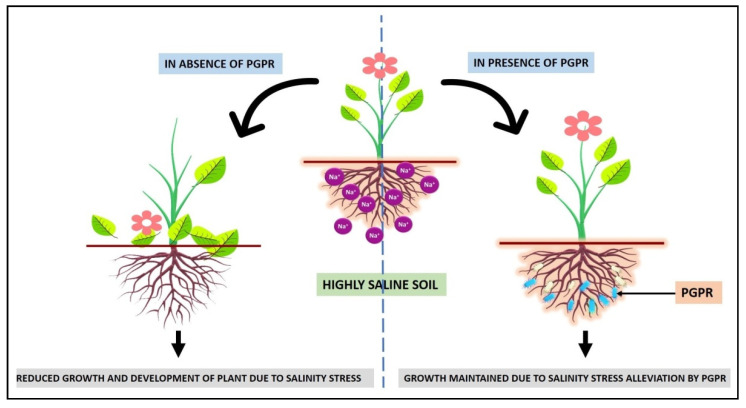
Effect of salinity on plants with and without the association of PGPR.

**Table 1 ijms-22-11461-t001:** PGPR mediated induction of salinity stress in plants.

Bacteria Used	Plant	Salt Treatment	Mode of Treatment	Beneficial Effects	References
*Staphylococcus kloosii, Kocuria erythromyxa*	*Raphanus sativus*	80 mM	Seed	Increased fresh and dry root weight, fresh and dry shoot weight, chlorophyll content, plant nutrient element contents of leaves	Yildirim et al. [232]
*Pseudomonas fluorescens*	*Zea mays*	15 dS m^−1^	Seed	Increased root length, plant height, phosphorous uptake, nitrogen uptake with enhanced grain yield	Nadeem et al. [233]
*Pseudomonas putida*	*Gossypium hirsutum*	Secondary salinized soil type	Seed	Increased germination rate, fresh and dry weight, plant height, K^+^ concentration	Yao et al. [234]
*Enterobacter aerogens*, *Bacillus brevis*	*Solanum melongena*	25 mM	Seedling	Increased shoot fresh and dry weight, root dry weight, uptake of N, P and K	Abd El-Azeem et al. [235]
*Bacillus subtilis*, *Pseudomonas fluorescens*	*Raphanus sativus*	75 mM and 150 mM	Seed	Increase fresh and dry root mass, fresh and dry shoot mass, fresh and dry leaf mass, chlorophyll content, carotenoid content, total photosynthetic pigment contents with improved N and P nutrition in plants	Mohamed and Gomaa [236]
*Bacillus aryabhattai, Brevibacterium epidermis, Micrococcus yunnanensis*	*Capsicum annum*	150 mM	Seedling	Increased dry root weight, root length, dry shoot weight, shoot length, increased ACS activity with decreased ethylene synthesis	Siddikee et al. [237]
*Burkholderia cepacian**, Promicromonospora* sp., *Acinetobacter calcoaceticus*	*Cucumis sativus*	120 mM	Seedling	Significantly higher biomass under salinity stress, downregulated ABA compared with control plants, while salicylic acid and gibberellin GA_4_ contents were increased	Kang et al. [238]
*Pseudomonas putida*	*Solanum lycopersicum*	90 mM	Seed	Increased shoot growth after 6 weeks in saline conditions, expression of Toc GTPase, a gene of the chloroplast protein import apparatus was upregulated, which may facilitate import of proteins involved as a part of stress response	Yan et al. [239]
*Bacillus amyloliquefaciens*	*Zea mays*	100 mM	Seedling	Increased chlorophyll content, total soluble sugar content and improved peroxidase and catalase activity, upregulation of genes *RBCS, RBCL* (encoding Ribulose 1,5-bisphosphate carboxylase/oxygenase subunits), H^+^-PPase (encoding H^+^ pumping pyrophosphatase), *HKT1, NHX1, NHX2* and *NHX3*	Chen et al. [240]
*Enterobacter* sp.	*Abelmoschus esculentus*	75 mM	Seedling	Enhanced salt tolerance, increased antioxidant enzymes and transcription of ROS pathway genes	Habib et al. [228]
*Herbaspirillum* sp.	*Brassica rapa*	150 mM	Seedling	Increased fresh and dry root weight, fresh and dry shoot weight	Lee et al. [241]
*Bacillus subtilis*	*Puccinellia tenuiflora*	200 mM	Seed	Reduced accumulation of Na^+^ ions	Niu et al. [242]
*Pantoea dispersa*	*Cicer arietinum*	40 mM and 60 mM	Seed	Increased biomass, number of pods and pod weight, seed number and seed weight, improved chlorophyll content and improved K^+^ uptake	Panwar et al. [243]
*Variovorax paradoxus*	*Pisum sativum*	70 mM and 130 mM	Seedling	Increased photosynthetic rate, electron transport with overall improvement in the plant biomass, increased root to shoot K^+^ flow and Na^+^ deposition in roots, thereby increasing K^+^/Na^+^ ratio in shoots	Wang et al. [179]
*Pseudomonas fluorescens*	*Zea mays*	150 mM	Seed	Improved root growth and promotion of root formation, release of IAA and protection against inhibitory effects of NaCl	Zerrouk et al. [244]
*Microbacterium oleivorans*, *Brevibacterium iodinum*, *Rickettsia massiliae*	*Capsicum annum*	200 mM	Seedling	Increased fresh and dry root weight, root length, fresh and dry shoot weight, shoot length, total chlorophyll content, total soluble sugar, proline content and antioxidant enzyme activity of APX, CAT and GPX.	Hahm et al. [245]
*Serratia liquefaciens*	*Zea mays*	80 mM and 160 mM	*Z. mays* plant	Increased growth and biomass yield, root length, shoot length, root fresh and dry weight, stem fresh and dry weight, chlorophyll content, carotenoid content, total soluble sugar and total soluble protein	El-Esawi et al. [246]
*Klebsiella* sp.	*Avena sativa*	100 mM	Seedling	Increased fresh and dry root weight, root length, fresh and dry shoot weight, enhanced biomass with high chlorophyll content	Sapre et al. [247]
*Enterobacter* sp.	*Oryza sativa*	150 mM	Seedling	Increased germination, fresh and dry root weight, root length, fresh and dry shoot weight, shoot length, chlorophyll content	Sarkar et al. [248]
*Burkholderia* sp.	*Oryza sativa*	185 mM	Seedling	Increased fresh and dry root weight, number of lateral branching roots and root length, fresh and dry shoot weight, enhanced seed germination, chlorophyll content	Sarkar et al. [249]
Consortium of *Aneurinibacillus aneurinilyticus* and *Paenibacillus* sp.	*Phaseolus vulgaris*	25 mM	Seed	Increased shoot length, root length with chlorophyll content	Gupta and Pandey [199]
*Pseudomonas putida*	*Capsicum annuum*	150 mM and 300 mM	Seedling	Increased fresh and dry root weight, fresh and dry shoot weight, nitrogen and phosphorous accumulation	He et al. [250]
*Leclercia adecarboxylata*	*Solanum lycopersicum*	120 mM	Seedling	Increased shoot length, stem diameter, shoot weight, root weight, chlorophyll fluorescence, sugar and amino acid synthesis	Kang et al. [251]
*Pseudomonas plecoglossicida*	*Zea mays*	150 mM	Seed	Increased root length, stem weight, stem height, fresh and dry weight of plant, chlorophyll content and total carbohydrate content	Zerrouk et al. [252]
*Azospirilum lipoferum, Azobacter chroococcum*	*Zea mays*	100 mM	Seed	Enhanced seedling leaf area, increased fresh and dry weight, chlorophyll and carotenoid content, total soluble sugar content and total soluble protein content	Latef et al. [253]
*Stenotrophomonas maltophilia*	*Arachis hypogea*	100 mM	Plantlets	Increased shoot length, fresh and dry plant weight and improved total chlorophyll content	Alexander et al. [254]
*Kocuria rhizophila*	*Zea mays*	100 mM and 200 mM	Seed	Increased root length, root dry weight, shoot height, shoot dry weight, chlorophyll content, soluble sugar content	Li et al. [255]
*Pseudomonas aeruginosa, P. resinovorans*	*Eleusine coracana*	350 mM	Seeds	Increased germination, vigor index, root length, shoot length, improved number of spikelets	Mahadik et al. [256]
*Bacillus safensis*	*Zea mays*	100 mM	Seedling	Increased root length, shoot length, fresh and dry weight of plant, number of leaves, chlorophyll and carotenoid content and total soluble sugar content	Misra and Chauhan [257]
*Sphingobacterium* sp.	*Lycopersicum esculentum*	200 mM	Seed	Enhanced plant biomass, root length, and shoot length, production of IAA and siderophores, phosphate solubilization	Vaishnav et al. [258]
*Pseudomonas migulae*	*Camelina sativa*	192 and 213 mM	Soil	Reduced the decline in shoot length, shoot weight and photosynthetic capacity, negatively affected ethylene signaling, auxin and JA biosynthesis and signaling, and positive effect on the regulation of genes in GA signaling	Heydarian et al. [72]

## Data Availability

The data presented in this study are available in this manuscript.

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
