# Peer review of "Plants Saline Environment in Perception with Rhizosphere Bacteria Containing 1-Aminocyclopropane-1-Carboxylate Deaminase"

_ijms, 2021, doi:10.3390/ijms222111461_

Round 1
Reviewer 1 Report
The presented manuscript deals with the issue of salinity and its effect not only on plants but also on microorganisms found in the rhizosphere. This is a very current issue, as salinity stress together with water deficit are the two most important stressors that affect the quality and level of production. The manuscript is conceived as a literature review, where attention is focused not only on the general characteristics of stressor and salinity, but especially on the effect of salinity on plant metabolism, hormonal activity, defense reactions, etc. This is a comprehensive overview of the issue within our current scientific knowledge. The manuscript is written very carefully, with considerable educational potential. It is possible only to the detriment that the text is not supplemented by a large number of pictures, diagrams. In my opinion, it would be appropriate to add this. The manuscript in this way has considerable potential for citation and use not only in teaching, but also in the direction of further research in this area.
Author Response
Authors’ response to the “REVIEWER 1” comments
We profusely thank the reviewers for their constructive comments. Herewith we are submitting the revised manuscript following incorporation of all the suggestions as indicated by the reviewers. We hope the reviewers agree with the corrections incorporated. In the revised manuscript, the questions raised by editor/ reviewers’ have been addressed and the changes are made in red colour.
Comment No. 1: The manuscript is written very carefully, with considerable educational potential. It is possible only to the detriment that the text is not supplemented by a large number of pictures, diagrams. In my opinion, it would be appropriate to add this. The manuscript in this way has considerable potential for citation and use not only in teaching, but also in the direction of further research in this area.
Response: We greatly acknowledge the reviewers thought in the manuscript. But we would like to state that as the manuscript is already included eight figures, we feel it is sufficient. All the figures have been depicted where and when necessary in the review. Hope reviewer agrees with this.

Reviewer 2 Report
This seems to be a well-conducted paper and to have clear, and informative results. The authors analyze relationships among changes in vegetation, climate, and land use.
I have no critical comments. The authors' views are not in line with many studies, however, the paper is innovative and can provoke future discussions. The structure of the paper is logical and the results are well reproduced. The introduction and discussion are well organized. The results reported have not been published elsewhere. Conclusions are presented in an appropriate fashion and are supported by the data.
This manuscript is original and presents an interesting packet of information.
The paper requires MAJOR revisions to be acceptable for publication, I have a few minor suggestions which I believe will improve the manuscript.
My suggestions regarding this MS is to accept after revising with the following points:
- The paper is well introduced. The title reflects the complexity and main objectives of the paper. The literary structure of the introduction is good, but authors should better interpret their research hypothesis of the importance of this kind of research.
- Authors should include the missing information (research gaps and the significance of the research). For improving the introduction and discussion sections authors could involve also new aspects about regulation mechanisms
- I would like to advise adding new pieces of information about biochemical and molecular mechanisms.
- The understanding of mechanisms is in some parts limited, as it is restricted to papers that have a particular view and deliberately ignore alternatives, and do not present a balanced view of the evidence. But paper presents novel results. The discussion lacks a bit in-depth.
I would like to invite authors to discuss more eco-physiological aspects and molecular mechanisms using/reading new actual references:
- Cross-talk between nitric oxide, hydrogen peroxide and calcium in salt-stressed. Chenopodium quinoa Willd. at seed germination stage. Plant Physiology and Biochemistry, 2020, Vol. 154, 2020, 657-664 doi: https://doi.org/10.1016/j.plaphy.2020.07.022
- Differential Response of Sugar Beet to Long-Term Mild to Severe Salinity in a Soil–Pot Culture. Agriculture, 2019, 9, 223; doi:10.3390/agriculture9100223
- Sustainable wheat (Triticum aestivum ) production in saline fields: a review, Critical Reviews in Biotechnology, DOI:10.1080/07388551.2019.1654973
- Crucial Cell Signaling Compounds Crosstalk and Integrative Multi-Omics Techniques for Salinity Stress Tolerance in Plants. Front. Plant Sci. 12:670369. doi: 10.3389/fpls.2021.670369
- Arbuscular Mycorrhizal Fungi and Plant Growth-Promoting Rhizobacteria Enhance Soil Key Enzymes, Plant Growth, Seed Yield, and Qualitative Attributes of Guar. Agriculture 2021, 11, 194. https:// doi.org/10.3390/agriculture11030194
I can support publishing this paper with major modifications.
Author Response
Authors’ response to the “REVIEWER 2” comments
We profusely thank the reviewers for their constructive comments. Herewith we are submitting the revised manuscript following incorporation of all the suggestions as indicated by the reviewers. We hope the reviewers agree with the corrections incorporated. In the revised manuscript, the questions raised by editor/ reviewers’ have been addressed and the changes are made in red colour.
Comment No. 1: The paper is well introduced. The title reflects the complexity and main objectives of the paper. The literary structure of the introduction is good, but authors should better interpret their research hypothesis of the importance of this kind of research.
Response: As per the reviewer’s suggestion, better interpretation has been made in the revised manuscript on the literary structure.
Comment No. 2: Authors should include the missing information (research gaps and the significance of the research). For improving the introduction and discussion sections authors could involve also new aspects about regulation mechanisms.
Response: As per the reviewer’s suggestion, the research gaps and significance of the study are included along with biochemical and molecular mechanisms involved during the salinity stress in the revised manuscript.
Comment No. 3: I would like to advise adding new pieces of information about biochemical and molecular mechanisms.
Response: As per the reviewer’s suggestion, information related to biochemical and molecular mechanisms have been included in the review.
Comment No. 4: The understanding of mechanisms is in some parts limited, as it is restricted to papers that have a particular view and deliberately ignore alternatives, and do not present a balanced view of the evidence. But paper presents novel results. The discussion lacks a bit in-depth. I would like to invite authors to discuss more eco-physiological aspects and molecular mechanisms using/reading new actual references:
- Cross-talk between nitric oxide, hydrogen peroxide and calcium in salt-stressed. Chenopodium quinoa Willd. at seed germination stage. Plant Physiology and Biochemistry, 2020, Vol. 154, 2020, 657-664 doi: https://doi.org/10.1016/j.plaphy.2020.07.022
- Differential Response of Sugar Beet to Long-Term Mild to Severe Salinity in a Soil–Pot Culture. Agriculture, 2019, 9, 223; doi:10.3390/agriculture9100223
- Sustainable wheat (Triticum aestivum ) production in saline fields: a review, Critical Reviews in Biotechnology, DOI:10.1080/07388551.2019.1654973
- Crucial Cell Signaling Compounds Crosstalk and Integrative Multi-Omics Techniques for Salinity Stress Tolerance in Plants. Front. Plant Sci. 12:670369. doi: 10.3389/fpls.2021.670369
- Arbuscular Mycorrhizal Fungi and Plant Growth-Promoting Rhizobacteria Enhance Soil Key Enzymes, Plant Growth, Seed Yield, and Qualitative Attributes of Guar. Agriculture 2021, 11, 194. https:// doi.org/10.3390/agriculture11030194
Response: As per the reviewer’s suggestion, the listed references have been discussed in the review with due importance to the research developments in the salinity stress and references have been cited related to eco-physiological and molecular mechanisms in the revised manuscript.

Reviewer 3 Report
The review presented by the authors is devoted to the problems of soil salinization. This problem is so global that many articles and reviews are devoted to it. From the title of the review it follows that it is devoted primarily to the protective effect of rhizobacteria containing ACC against the effect of salinization. Although the authors presented in the review a large and very informative table "PGPR mediated induction of salinity stress in plants", however, the main emphasis was placed on the various effects of salinity and it addresses a wider range of problems associated with salinity. Perhaps the authors should correct the title of the review.
The review is well written and well structured. However, there are comments on fig. 3. Although AdoMet is a necessary substrate for methylation of 1 aminocyclopropane -1-carboxylate, this is not its only reaction, it is involved in the methylation of many classes of compounds, so it is better to skip the first step in the scheme. And in fig. 5 part of the diagram on the left, you can skip the whole part shown in fig. 3. This is a repeat. Although the name of the enzyme ribulose-1,5-bisphosphate carboxylase / oxygenase - Rubisco is used, it is desirable to give its full name. There are some small design notes. There are sometimes missing gaps in the review (250, 292, 501, 523). 656- this or the.
The review is worthy of being published.
Author Response
Authors’ response to the “REVIEWER 3” comments
We profusely thank the reviewers for their constructive comments. Herewith we are submitting the revised manuscript following incorporation of all the suggestions as indicated by the reviewers. We hope the reviewers agree with the corrections incorporated. In the revised manuscript, the questions raised by editor/ reviewers’ have been addressed and the changes are made in red colour.
Comment No. 1: Perhaps the authors should correct the title of the review.
Response: We would like to state here that as the title suits the special issue we would like to keep the title as it is. But we further state that some parts in the revised manuscript have been updated for the suitability of the same.
Comment No. 2: However, there are comments on fig. 3. Although AdoMet is a necessary substrate for methylation of 1 aminocyclopropane -1-carboxylate, this is not its only reaction, it is involved in the methylation of many classes of compounds, so it is better to skip the first step in the scheme.
Response: As per the suggestion, the first step in the scheme of figure 3 has been skipped in the revised manuscript.
Comment No. 3: And in fig. 5 part of the diagram on the left, you can skip the whole part shown in fig. 3. This is a repeat. Although the name of the enzyme ribulose-1,5-bisphosphate carboxylase/ oxygenase - Rubisco is used, it is desirable to give its full name. There are some small design notes.
Response: As per the suggestion, part of diagram on left of figure 5 has been skipped. The full name of enzyme, ribulose-1,5-bisphosphate carboxylase/ oxygenase is replaced for – Rubisco in the revised manuscript.
Comment No. 4: There are sometimes missing gaps in the review (250, 292, 501, 523). 656- this or the.
Response: We thank the reviewer for his constructive comments and all the corrections have been incorporated in the revised manuscript.

Round 2
Reviewer 2 Report
Dear authors,
Now the article is in good shape for acceptance to publish. The queries raised were well addressed and the authors revised the article in a worthwhile manner.
Author Response
Comment No. 1: Now the article is in good shape for acceptance to publish. The queries raised were well addressed and the authors revised the article in a worthwhile manner.
Response: We profusely thank the reviewer for accepting the revised manuscript.